# Health professional and patient views of a novel prognostic test for melanoma: A theoretically informed qualitative study

Jan Lecouturier[1]*, Helen Bosomworth[2], Marie Labus[3], Rob A. Ellis[4,5,6], Penny E. Lovat[4]

1 Population and Health Sciences Institute, Newcastle University, Newcastle upon Tyne, United Kingdom, 2 Institute of Neuroscience and Newcastle University Institute for Ageing, Newcastle University, Newcastle upon Tyne, United Kingdom, 3 Research and Enterprise Services, Faculty of Medical Sciences, Newcastle University, Newcastle upon Tyne, United Kingdom, 4 Precision Medicine, Translation and Clinical Research Institute, Newcastle University Centre for Cancer, Newcastle upon Tyne, United Kingdom, 5 South Tees Hospitals NHS Foundation Trust, The James Cook University Hospital, Middlesbrough, Cleveland, United Kingdom, 6 York Teaching Hospital NHS Foundation Trust, The York Hospital, York, North Yorkshire, United Kingdom

☯ These authors contributed equally to this work.
* jan.lecouturier@newcastle.ac.uk

**Data Availability Statement:** There are ethical restrictions on sharing the data as we do not have the consent of participants to share it beyond the study team. Requests can be made to the study

## Abstract

### Objectives

Cutaneous melanoma rates are steadily increasing. Up to 20% of patients diagnosed with AJCC Stage I/II melanomas will develop metastatic disease. To date there are no consistently reliable means to accurately identify truly high versus low-risk patient subpopulations. There is hence an urgent need for more accurate prediction of prognosis to determine appropriate clinical management. Validation of a novel prognostic test based on the immunohistochemical expression of two protein biomarkers in the epidermal microenvironment of primary melanomas was undertaken; loss of these biomarkers had previously been shown to be associated with a higher risk of recurrence or metastasis. A parallel qualitative study exploring secondary care health professional and patient views of the test was undertaken and this paper reports the perceived barriers and enablers to its implementation into the melanoma care pathway.

### Methods

Qualitative methods were employed drawing upon the Theoretical Domains Framework (TDF) in the exploration and analysis. An inductive-deductive analysis was performed, with all data coded using a thematic then TDF framework.

### Findings

20 dermatologists, plastic surgeons, cancer nurse specialists, oncologists and histopathologists participated. Nine TDF domains were relevant to all health professional groups and the 'Skills' and 'Beliefs about Capabilities' domains were relevant only to histopathologists.

sponsor's Data Protection Team at stees.dpo@nhs.
net who were named on the materials given to
participants regarding General Data Protection
Regulations.

**Funding:** This study was funded by the National
Institute for Health Research (NIHR) https://www.
nihr.ac.uk/ Invention for Innovation (i4i)
programme (II-LA-0417-20001) https://www.nihr.
ac.uk/explore-nihr/funding-programmes/invention-
for-innovation.htm. The views expressed in this
manuscript are those of the author(s) and not
necessarily those of the NIHR or the Department of
Health and Social Care. PL is the author who
received this award. At the time of the study ML
was CEO of AMLo Biosciences on a consultancy
arrangement with Newcastle University (the host
institution for the i4i grant). AMLo Biosciences
website www.amlo-biosciences.com. The funders
had no role in study design, data collection and
analysis, decision to publish, or preparation of the
manuscript.

**Competing interests:** I have read the journal's
policy and the authors of this manuscript have the
following competing interests: ML is employed
fulltime as the CEO of AMLo Biosciences. RAE, ML
and PL are shareholders in AMLo Biosciences. PL
is employed part-time as Chief Scientific Officer
with AMLo Biosciences. AMLo Biosciences hold
the patents for the test.

'Optimism' and 'Beliefs about consequences' were strong enablers particularly for clinicians. 'Environmental context and resources' (impact on pathology services) and 'Knowledge' (the need for robust evidence about the test reliability) were the main perceived barriers. 19 patients and one carer were interviewed. For the patients eight domains were relevant. ('Knowledge', 'Emotions', 'Beliefs about consequences', 'Social Role and identity', 'Behavioural regulation', 'Memory, attention and decision processes', 'Reinforcement' and 'Skills'). The consequences of the implementation of the test were reassurance about future risk, changes to the follow-up pathway on which there were mixed views, and the need to ensure they maintained self-surveillance (Beliefs about consequences). The test was acceptable to all patient interviewees but the resultant changes to management would need to be supported by mechanisms for fast-track back into the clinic, further information on self-surveillance and clear management plans at the time the result is conveyed (Behavioural regulation).

## Conclusions

Health professionals and patients perceived positive consequences—for patients and for health services—of adopting the test. However, its implementation would require exploration of the resource implications for pathology services, psychological support for patients with a high-risk test result and mechanisms to reassure and support patients should the test lead to reduced frequency or duration of follow-up. Exploring implementation at an early stage with health professionals presented challenges related to the provision of specific details of the test and its validation.

## Introduction

Cutaneous melanoma is the UK's fifth most common cause of cancer with approximately 16,200 new melanoma skin cancer cases diagnosed in the UK every year [1]. Incidence has continued to rise over the last sixty years, particularly in younger people, and this trend is set to continue [2].

In the UK, patients are likely to consult a general practitioner (GP) with a mole or lesion they are concerned about (Fig 1). The GP will examine the skin and if a melanoma is suspected, refer the patient to a secondary care clinic for a clinical diagnosis to be made by a specialist (usually a dermatologist or plastic surgeon). The suspected melanoma is removed by surgical excision and diagnosis confirmed by microscopic examination of the tissue by a histopathologist, with internationally accepted criteria (tumour depth, Breslow depth, degree of tumour invasion/spread) within the 8th edition of the American Joint Committee on Cancer (AJCC) staging system [3] used to guide subsequent treatment and prognosis. Patients with an AJCC Stage I or II melanoma—i.e. those which have not metastasized—may undergo a further Wide Local Excision (WLE) to ensure a margin of nearby skin around the melanoma and the removal of all cancerous tissue and minimise the risk of reoccurrence. Some patients with Stage IB and Stage II melanoma will be offered a sentinel lymph node biopsy (SNB) to determine whether there has been any spread of the melanoma cells into the lymph nodes. This procedure involves injecting a radioactive dye into the tissue around the site of the melanoma and removal of the initial lymph node(s) for pathological assessment. Whilst a SNB can offer further information about the potential of melanoma progression this is not definitive, and the

## The melanoma care pathway

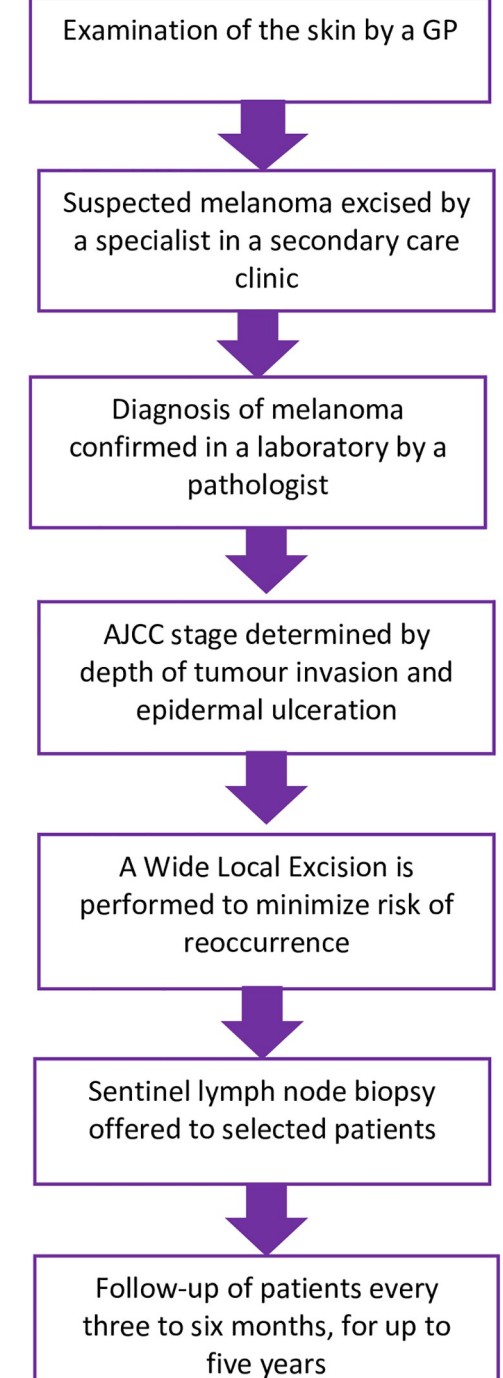

**Fig 1. Melanoma care pathway.**

procedure is invasive. Following a Stage I or II diagnosis, the National Guidelines for melanoma recommend follow-up by a specialist (dermatologist, plastic surgeon or cancer nurse specialist) every three to six months, for up to five years [4]. At follow-up the patient is examined for signs of either reoccurrence, by checking the original melanoma site and lymph nodes, or signs of any new primary melanomas. Additional adjuvant treatments to try and prevent progression of the melanoma can be offered in situations where there is metastatic melanoma detected at SNB or at a later stage in the lymph nodes.

Approximately 10% of patients with Stage I and 20% with Stage II melanomas will go on to develop metastatic disease with an extremely poor prognosis [3]. Whilst able to stage tumours based on tumour size, depth and degree of spread, current AJCC staging criteria do not distinguish between truly high and low risk Stage I/II melanomas and currently there are no other consistent ways to accurately determine an individual's risk of disease progression. There is hence an urgent need for more accurate prediction of prognosis to determine future management such as counselling and follow-up or access to clinical trials and adjuvant therapy [5, 6].

Prognostic biomarkers for determining disease risk are a promising area of research. Funding was secured to validate a novel prognostic test developed to accurately identify and distinguish between early AJCC stage melanomas that are genuinely low risk of disease progression from those that are high-risk, with greater sensitivity and specificity than any other prognostic melanoma tests in development.

The combined immunohistochemical expression of two protein biomarkers, AMBRA1 and Loricrin in the epidermis overlying a primary melanoma as compared with expression in the normal epidermis in the same excision biopsy, has been identified as a novel prognostic biomarker for early AJCC stage I melanomas [7]. Based on the presence (low risk) or absence (at risk) of these protein, and with a high negative predictive value, this test allows the identification of genuinely low risk tumour subsets and hence the potential for reduced patient clinical follow up. Those patients with tumours in which epidermal AMBRA1 and loricrin is lost remain at risk of recurrence or spread and their follow up should be continued in line with current surveillance guidelines. Clinical utility analysis of this novel prognostic in a discovery cohort of 379 AJCC stage I melanomas has revealed a NPV of 98.3%, a PPV of 13.6% and an assay sensitivity and specificity of 82.6% and 66% respectively [7]. The test is performed on the primary tumour biopsy taken as part of routine clinical practice, with sections of the tumour stained in a diagnostic cellular pathology laboratory using standard automated immunohistochemistry, underpinning the lack for any additional clinical procedures required for the patient.

There is a dearth of studies in the cancer literature pertaining to both health professional and patient barriers and enablers to the implementation of novel prognostic tests, particularly at an early stage in the validation of a technology. We explored the views of health professionals and patients about this novel test and its potential implementation into the current NHS melanoma pathway. This qualitative component was conducted in parallel to the validation work. This paper reports the barriers and enablers to the implementation of this novel test in the melanoma care pathway from the perspectives of health professionals working in secondary care and patients with melanoma.

## Methods

### Recruitment and data collection

This was a qualitative study. Single semi-structured telephone interviews were conducted with health professionals involved in the melanoma diagnostic and care pathway and patients diagnosed with melanoma. Interviews were conducted by an experienced qualitative researcher

(JL) who was a co-investigator though not involved in the development or validation of the novel test. The researcher was not known to participants prior to being contacted about the study.

A favourable ethical opinion was obtained from East of England—Cambridge East Research Ethics Committee on 11th February 2019 (19/EE/0018).

The adoption and implementation of new technologies requires a change in the behaviour of the intended user and there is a need to understand the determinants of the behaviour. This inquiry was informed, and the analysis guided, by the Theoretical Domains Framework (TDF) [8, 9]. The TDF is based upon 33 theories of behaviour and behaviour change clustered into 14 domains and provides a framework or 'lens through which to view the cognitive, affective, social and environmental influences on behaviour' (page 2) [10]. It has been employed in studies exploring the implementation of new practices, for example an intervention for general practitioners about human papillomavirus infection, vaccination and testing [11] and the operating room black box [12].

Study information for both health professionals and patients was developed and described the aims of the study, what was required of participants. the names of the core research team, and ethical approvals. A small number of patients from a local Melanoma Support Group reviewed the patient information materials and recommended changes to improve comprehension and readability.

A purposive sample of UK secondary care health professionals involved in the melanoma diagnostic and care pathway were recruited via professional networks. Study information was circulated through Melanoma Focus (https://melanomafocus.org/members/apply-or-pay-for-membership/) and the Northern Cancer Alliance Skin Cancer Speciality Group https://northerncanceralliance.nhs.uk/advisory_group/skin-expert-advisory-group/ (the lead of which is a co-applicant on the study). In the latter, members were asked to cascade the information to relevant others outside of the northern region. Anyone connected to the study or the development of the novel test was excluded. Names were passed to the researcher who approached individuals by email to take part in an interview, or those who were interested could contact the researcher directly. This method of identifying potential participants resulted in the recruitment of interviewees from a range of NHS Trusts in England and Scotland. This enabled the exploration of the perspectives of those working in NHS Trusts with different care pathways.

A purposive sample of patients with melanoma, a combination of those diagnosed within the last six months and up to five years, were recruited from two NHS hospital trusts. They were approached by the responsible clinician or research nurse and given a study pack and then followed up to ask if they would be happy to be approached for interview. The details of those who agreed to discuss participation in the study were sent to the researcher.

Data collection for both patients with melanoma and health professionals ended when data saturation was achieved, and no new themes were identified.

Brief information about the novel test (Fig 2) was sent to interviewees just prior to the interview or read out at the time. This was written in lay terms for use with a range of health professionals and patients. We believed that health professionals and patients had the requisite experience and understanding of the care pathway to be able to provide valid comments on barriers/enablers of implementing this test despite the fact it was a test not currently used in clinical practice. Participants provided verbal consent to take part in this study at the time of the interview. Written consent was not obtained as the data were collected by telephone interviews. Participant consent was documented by the researcher (JL) on a paper version of the consent form, and the consent interaction was digitally audio-recorded. The researcher read each statement and the interviewee stated whether they understood/agreed. On the paper

When someone is diagnosed with melanoma it is difficult for health professionals to know with any certainty whether the tumour will return or spread. The chance of their tumour returning or spreading is calculated based on certain features of the tumour. However, out of every 100 tumours classified as low risk, ten will return or spread. The (name) test aims to accurately identify melanomas that are genuinely low-risk and those that are genuinely high-risk.

The (name) test is performed on the tumour biopsy that is taken as part of routine clinical practice and does not require any additional appointments or biopsies. In the laboratory, sections of the tumour are fixed onto microscope slides and stained for two proteins. These slides are then loaded into a scanner which magnifies the sections allowing a pathologist to look at the slides to see whether the two proteins of interest are present within the skin overlying the tumour. If the proteins are present in the skin then the melanoma is classified as low-risk and if the proteins are lost in the skin then the melanoma is high-risk and patients will need to be monitored further.

It is proposed that patients will be informed of the result of the (name) test at their normal follow-up appointment. The intention is that the results will help clinicians and patients to decide on the most appropriate management going forward.

**Fig 2. Information about the test provided to interviewees.**

version, the researcher recorded their own initials in a box by each statement to indicate that the interviewee understood/agreed. The ethics committee agreed this procedure. A topic guide designed to explore the TDF domains was used. The guide for the patient interviews was developed in collaboration with a local Melanoma Support Group and for the health professionals with the wider study team. The interviews were recorded digitally with the permission of the interviewee.

## Analysis

The interview sound-files were transcribed verbatim and NVivo software was used as a management tool. A thematic analysis was conducted which involved a process of familiarisation with the data to develop health professional and patient coding frameworks (Table 1) [13]. The data were coded firstly using this framework and then the TDF domains. This inductive-deductive approach avoided the loss of themes elicited from the data unrelated to the TDF domains [14]. Two members of the research team (JL, HB) coded the data using the TDF framework and checked for consistency. Any discrepancies were discussed and resolved through consensus.

**Table 1. Steps in inductive data analysis—Adapted from [13].**

| |
|---|
| **Step 1: Familiarisation with data**—reading, re-reading transcripts and listening to soundfiles from interviews/focus groups |
| **Step 2: Generate initial codes**—systematically record features that are interesting across the data |
| **Step 3: Identify themes**—coded extracts are sorted into overarching themes and subthemes |
| **Step 4: Review of themes**—themes are combined, refined, redefined, or separated and a map or framework devised |
| **Step 5: Defining and naming themes**—refinement of the themes and sub-themes and the addition of concise working definitions of each theme |

## Findings

Twenty health professionals were interviewed (JL) between June 2019 and March 2020. These included: dermatologists (5), cancer nurse specialists (3), plastic surgeons (4), oncologists (4) and histopathologists (4) from 12 UK NHS hospital trusts. It was not possible to determine how many health professionals received study information through the two professional groups and therefore it is difficult to calculate a response rate or reasons for not participating in the study. A few histopathologists emailed to say there was too little information provided about the test to enable them to participate in an interview. Twenty people (19 patients diagnosed with Stage I or II melanoma and one carer) were interviewed between July 2019 and December 2020. Six had been diagnosed for less than 12 months, nine for one-two years and five for three to four years. The interview duration for health professionals ranged from 20–60 minutes and for patients 50–60 minutes.

The health professional data, including that coded into the TDF domains, are presented in two main sections: views of the novel test including the perceived outcomes, individual incentives and motivation to use it; and perceived issues around the future implementation of the test. The patient data are presented in three sections: views of the novel test, what the test means for patients, and structures and behaviours to support test-based decision-making. Quotations illustrative of the TDF domains for health professional and patient data are provided in Tables 2 & 3.

### Health professional views of the novel prognostic test

At the time of the present study the novel test described within this manuscript was undergoing final clinical validation. The majority of interviewees had no prior knowledge of the novel test other than what was shared as part of the interview, though a few had attended presentations describing the development of the test at national/international meetings including the British Society for Investigative Dermatology, and Melanoma Focus. All were asked for their general views of the test as part of the melanoma care pathway. Regardless of speciality, everyone was very positive about the test and at a general level its perceived potential was to identify those at risk of recurrence or spread and prevent that happening (Goals). One interviewee said the test would be 'invaluable' if proven to provide better prognostic information (Emotion). The test was welcomed as it was acknowledged they currently had little to offer this group of patients in terms of information on future risk (Optimism).

There were no specific issues raised about the test fitting into the care pathway as clinicians already discuss the risk of spread or recurrence with patients (social/professional role and identity). As the test is new, however, some talked about the need to continue to draw upon the current criteria in determining prognosis.

**Perceived impact of the prognostic test.** There were four perceived positive consequences of introducing the test into the care pathway. These related primarily to the potential to tailor

**Table 2. Theoretical Domains Framework and illustrative clinician quotations.**

| TDF Domain | |
|---|---|
| Knowledge | 'You would need to demonstrate the sensitivity and specificity of your test. It's all about giving people confidence that it is reliable and accurate and . . .that we can actually discharge a whole bunch of patients.' Int 306 Oncologist |
| Skills | 'I don't think that will be a big issue for us. [. . .] so all the pathologists here are able to report this immunohistochemistry and implementing and reporting it's not a big deal for us.' Int 310 Histopathologist |
| Social/professional role and identity | 'I would still probably need to use (test) in conjunction with other criteria I suppose but it depends how reliable it is, if it can be shown to be a really robust test, then that's ideal.' Int 315 Dermatologist |
| Beliefs about capabilities | 'It's more work, but actually, if we are selective on the cases that we do it on . . . only the ones that have a sort of morphological diagnosis of melanoma then there will be a significant number of cases but I'm sure it will be do-able.' Int 312 Histopathologist |
| Optimism | 'It's a fantastic idea and I would love it to work, . . . but I've been here before and everybody's said this is the best thing since slice bread and it doesn't live up to it. So you have to be pragmatic and say I'm very interested. . .There's enough information here to make it an exciting concept and test investigation to take it further but we can't do anything formally with it until we have the evidence out there.' Int 318 Plastic Surgeon |
| Beliefs about consequences—Perceived outcomes of test | 'So if you've got a test which can accurately define those patients at high or low risk, then that has significant implications for a whole range of things, even do you need to have a sentinel node biopsy, do you need to have 5 years follow-up, should you even be considered for adjuvant therapy even if your Breslow is 4 millimetres? Those are potential things.' Int 318 Plastic Surgeon |
| Reinforcement—Perceived incentives to adoption of test | 'If they thought it would be helpful in terms of lower risk patients. Even if they weren't discharged and seen every six months instead of every three, that would improve—it would free up an appointment . . . and with the rise in numbers of skin cancers something that you see is an awful lot of melanoma follow-ups.' Int 303 Dermatologist |
| Goals | 'Well it's relevant in that I'm interested in reducing the number of patients who end up having advanced disease. So it is very important and in terms of the resources used for surveillance etcetera so. . . Yeah. In the bigger picture, yes, I'm definitely interested in this.' Int 306 Oncologist |
| Memory, attention and decision processes | 'From a clinical perspective, if it's demonstrated to be useful in affecting your follow up and your potential management of patients then I think if it was accepted that would be a no-brainer. I think it would be used provided it was quick and useful and prevented as I said, the morbidity of anything that's more invasive.' Int 305 Plastic Surgeon |
| Environmental context and resources | 'Well it would be more time for (pathologists) because it would be an additional test which they don't currently do. That would be a problem for them because at the minute they're struggling.' ' Int 303 Dermatologist |
| Social influences | 'I think it would be a discussion with—because obviously it would have to go along with the clinical teams feeling comfortable . . . so I think it would probably be a discussion for us to have as a multidisciplinary team and we do have annual review meetings and a structure around making changes to our current way of doing things.' Int 309 Histopathologist |
| Emotion | 'I think it's invaluable if you can get it to work and prove it.' Int 313 Plastic Surgeon |

patient care but also to provide greater reassurance to patients (Beliefs about consequences). Tailored care encompassed elements of both the surveillance and treatment pathway.

*Surveillance pathway*. Most interviewees suggested the test could enable a more tailored approach to surveillance and the potential for different patient pathways depending on the test

**Table 3. Theoretical Domains Framework and illustrative patient quotations.**

| TDF Domain | |
|---|---|
| Knowledge | *'Because the follow ups cost money, at some time or other you've got to say 'we've proven this* (test) *now. We're not going to do it anymore.' I think that's how scientific development works doesn't it*? **Patient 01—Female– 14 months since diagnosis** |
| | *'I would be delighted to have it as an additional indicator, but I wouldn't consider it to be 100% accurate and maybe if like (clinician) was like, 'Look it's 90% accurate.' I'd be like it's fine but I mean is that fair or should we be considering it to be 100% accurate? Does it not need a little bit more data?'* **Patient 09 –Male– 3 years since diagnosis** |
| Reinforcement—Perceived incentives to adoption of test | *'Obviously it will save—if it's successful. If it's proven to be accurate it will save a lot of hospital time you know, both for the consultants and the patient so it's a good thing.* **Patient 13 –Female– 3 years since diagnosis** |
| Emotion | *'I think the fact that there is a test or there will be a test to determine whether you are low risk or high risk would be a bit of a relief in itself to be honest with you. That's just me personally anyway.'* **Patient 15 –Male– 11 months since diagnosis** |
| | *Are we more relying on the test than we are the patient and 'Go away? We don't need to see you anymore,' type of situation and then the odd one slips through the net. . . .It's a difficult question that because yes it would be better because the resources aren't being tied up so much and they can see more people and there is longer time but you know, you'd be a bit frightened that it becomes economical you know'.* **Patient 06 –Male– 5 years since diagnosis** |
| Memory, attention and decision processes | *'My own personal inclination would be that it should be compulsory for the simple reason that if it enables them to identify you as a high risk patient, I mean in the words of the hand out* (information about the novel test), *'They will need to be monitored further. . .'.* **Patient 10 –Male– 2 years since diagnosis** |
| Beliefs about consequences—Perceived outcomes of test | **Certainty and reassurance** |
| | *'One of the questions I was asking at the time.. was "How worried should I actually be about this?" and the answer was really diplomatic "We don't know. It's good that we've had it removed now, it's very small and there's no evidence that it's spread anywhere else or that it will come back necessarily but obviously we don't really know. It depends," but I think for me I would have quite liked—if there was a way to say, "Well actually, you're more likely to have problems with this in the future" or "Actually you're not particularly high risk to having additional problems in the future." That would have been welcome information.'* **Patient 02 –Male– 10 months since diagnosis** |
| | *'In my own case if I'd been told I was very low risk . . . I think that I would have been a little bit more reassured.'* **Patient 03 –Male– 2 years since diagnosis** |
| | **Impact on clinical decisions and patients' behaviour** |
| | *'I would rather know that I didn't need those three-monthly, six-monthly or whatever it would be.'* **Patient 08 –Male– 3 years since diagnosis** |
| | *'I tend to be a tiny bit glass half empty my.. thought is I can relax for six months because I know somebody will be looking at my skin and they've always been really good. If you notice any unusual or you know you can just call and make an appointment so that's what's been reassuring but I do worry a bit . . .because at the moment they're saying is after five years if nothing unusual has occurred on your skin then that's it we won't monitor you any more.'* **Patient 18 –Female– 4 years since diagnosis**. |
| | *'From my perspective, I think I would like to know whether there is a greater or lesser chance of it returning. I think that probably helps to make you slightly more vigilant to be truthful.'* **Patient 04- Female– 6 months since diagnosis** |

(*Continued*)

**Table 3.** (Continued)

| TDF Domain | |
|---|---|
| Behavioural regulation | *'If somebody like me is happy to be taken off with a waiver that if anything should happen you can be short-circuited or you'd contact a person to be reviewed more quickly. That might be an option.'* **Patient 07 –Female– 2 years since diagnosis** |
| | *'Maybe if there was like a phone number. If you noticed some change and you were worried—instead of going to your GP and making an appointment to see them . . . if there was a contact at the hospital that you can ring and ask their advice and they could actually get you get you in quicker to take a look maybe as like a safety net.'* **Patient 15 –Male– 11 months since diagnosis** |
| | *'If you could take away the follow up appointments but then still have open access for a year that might be. so that it's like on this last occasion when I went for a check-up I've got a couple of moles that I'd got queries about, I could phone them up and say, 'Can I have an appointment for you to check these out?' If they could do that still you know to sort of—well either identify anything or to put your mind at rest.'* **Patient 17 –Female– 9 months since diagnosis** |
| | *'If I'm high risk then what does that mean? I mean have I got days, months, years. . .? What? You know. I wouldn't want them to come back to me and say, 'Ah we've done the test. It's high risk.' Well I need them to back that up and unless they can do that I don't want the test, then I don't want to know.'* **Patient 05 –Female—4 months since diagnosis** |

result. The suggested changes ranged from immediate discharge to a reduction in the duration and, or frequency, of follow-up for those whose test indicates they are at low risk. The two main benefits to limiting follow-up attendance were more efficient use of resources by directing care to where it is needed and reducing the negative physical and psychological consequences of frequent and longer-term follow-up.

When asked whether a shorter or less frequent follow-up may raise anxiety for patients deemed low risk, the responses were mixed. The cancer nurse specialists in particular said patients may be more anxious, and the follow-up provides an opportunity for them to seek reassurance. Others said being discharged could be viewed positively by patients and reinforce the test information that their future risk is low; and with greater certainty about risk, being discharged will be acceptable to patients.

Those who recognised the potential to limit the follow-up or early discharge of low-risk patients said this would have to include a clear fast track process back into the system should the patient have any concerns.

In contrast, one interviewee did not consider a change to the follow-up pathway a positive outcome of the test. They said the one-year follow-up provided an opportunity for the clinical team to check whether patients understood and could carry out self-surveillance competently. This gave a certain level of reassurance to the health professionals.

*Sentinel node biopsy and lymph node dissection.* The second perceived consequence of adopting the test, primarily from the plastic surgeons, was to inform decisions about SNB and subsequent regional lymph node dissection. It was said the number of SNBs has increased considerably and '50% of all melanomas are considered for a sentinel node biopsy'. One interviewee said with the requisite levels of sensitivity and specificity the test could replace SNBs, the current gold standard. Another perceived consequence of the test, with the potential to bypass the SNB, was to offer new immunotherapies to those at high risk of metastases, thereby avoiding the invasive SNB procedure and the progression to lymph node dissection unnecessarily.

There were perceived benefits for patients of increased clinician certainty about lymph node dissection based on the test finding. These were the avoidance of surgery where it is unnecessary which 'would save the patient the morbidity and the stress and the time for the surgery' but also to improve the care of those where it is indicated.

*Adjuvant therapy*. The third potential outcome related to decisions about who should be offered adjuvant therapy. This was mentioned by oncologists, dermatologists and plastic surgeons. Interviewees stated in current practice those with a positive SNB and who are deemed to be fit enough are offered adjuvant therapy. The novel test could provide more clear-cut information on risk and guide decision-making about adjuvant therapy. This would avoid those at low risk of recurrence, estimated by one interviewee to be 50%, undergoing unnecessary treatment.

Some other potential outcomes of the test related to clinical trial eligibility criteria and also —although the interviews were focused on Stage I and II melanomas—its use for later-stage melanoma in decision-making about adjuvant therapy.

*Reassurance to patients*. The final perceived outcome was to provide more clearly defined prognostic information and greater reassurance to patients. Although it was acknowledged the test may raise anxiety for patients deemed at a higher risk of recurrence or spread, having this more clearly defined prognostic information could be of benefit to the larger proportion likely to be at a lower risk.

One interviewee also said having a formal prognostic test with 'a defined sort of statistic to it' would provide greater reassurance to patients. Another said even those patients whose test indicates they are low risk will remain anxious and the only reassurance would be to keep them under surveillance.

**Incentives.** The main incentive (reinforcement) for interviewees in a clinical role for adopting the test was the potential to increase capacity for clinicians. The most commonly mentioned incentive, based on changes to the follow-up pathway, was freeing up out-patient clinic time for dermatologists, specialist nurses and plastic surgeons.

If the novel test is proven to be effective in identifying risk, one interviewee stated the differences it would make to guiding decision-making for SNB and adjuvant therapies could lead to significant cost-savings. The potential for cost savings to the NHS through a reduction in the number of patients attending for follow-up was also raised.

There were no specific incentives identified by the histopathologists. This would be an additional laboratory test and would require extra resources in time and costs.

**Implementation.** *The need for evidence*. When exploring barriers to its implementation, as might be expected, the level of evidence about the ability of the test to accurately measure risk for Stage I and II melanomas was cited as a major factor impacting on its adoption. Apart from a small number who were unable to comment without more information about the test, most were clear about the evidence required for it to be implemented into the melanoma care pathway (Knowledge). When asked what evidence was required, suggestions were made about data collected retrospectively and prospectively and approval from The National Institute for Health and Care Excellence (NICE).

There were a number of questions about the validation study being conducted at the time. These included the age and number of samples, where they were obtained from—UK or wider—and whether the data would be sufficiently robust to change practice. Potential disadvantages of retrospective studies were stated, namely changes in melanoma management since the samples were collected and the potential for bias. Views were divided on whether a further prospective study was needed. Some interviewees commented that evidence from the current validation may be adequate for adoption if it had a robust design. A few said a further retrospective study could be conducted in-house in their own NHS Trust.

Others thought a prospective study should follow the current validation to reassure clinicians. Some who mentioned the need for a prospective study said this could be conducted in-house where patients would have the test and continue to be followed up as per the current guidelines.

Some commented the potential changes to the care pathway made on the strength of the test, required a paradigm shift and this would be a challenge. There were potential risks as a result of changing practice, for patients and for health professionals.

NICE approval was raised, though this was not considered a stumbling block for the adoption of the test. One interviewee said NICE approval would impact on whether their pathology lab could undertake the test themselves; and the implications of it being conducted elsewhere were delays in clinicians and patients receiving the test result.

*Social influences*. Social influence, for example the influence of colleagues on adoption of the test, was not a major factor. This domain was not identified in the interviews with dermatologists, oncologists or plastic surgeons. However, it was mentioned by some of the histopathologists and cancer nurse specialists who considered test adoption a group rather than individual decision.

*Environmental context and resources*. Another factor said to influence implementation of the test was its impact on pathology staff. Overwhelmingly health professionals said their pathology services were overstretched. However, although a number of interviewees considered the time taken to conduct an additional test would be problematic for pathologists, this view was not always shared by the histopathologists interviewed.

Some clinician interviewees indicated the potential benefits of the test outweighed the costs of the additional time needed for pathology and this is just one extra in a range of tests conducted. Features of the test, such as ease of use for pathologists, how much it costs and whether it will provide clear cut information on risk were mentioned as ways to mitigate any extra time involved. One other suggestion was to use the test only where malignancy is detected rather than all suspected melanoma samples. A few interviewees said the novel test may replace some of those currently conducted and therefore offset any extra time needed. It was also thought possible that the savings incurred in other departments through the use of the test may filter through to the pathologists.

*Skills and beliefs about capabilities*. Based on the information provided about the novel test, the view was that histopathologists would have the skills to conduct the test without any extensive training (Skills). Also some histopathologists considered the test to be merely one extra to add to the range they currently conduct (Beliefs about capabilities). However, expanding upon the point that the test may not always provide a clear-cut result, some histopathologists suggested a test protocol to guide its use, and define any grey areas and how they should be managed.

## Patients views of the novel prognostic test

Patient interviewees were asked about their experience of melanoma and the care they received including information on risks. Views of the test and its implications for melanoma treatment and management decision-making, were explored based on the TDF.

Most demonstrated knowledge of tests of this kind, pointing out that few are 100% accurate and that those offered through the health service have undergone rigorous testing. A few expressed the view that as it is new it should be used initially in conjunction with current prognostic measures until proven.

The main reinforcement for adopting the test was the impact it could have on the health services in alleviating the pressure of busy clinics on staff. Another was that this could improve

the care for those who are categorised as high risk who require more intensive follow-up or care.

Within the emotions domain, it was said it would be a 'relief' to have such a test. There were no concerns about the test per se as it is conducted on the original skin biopsy and requires no further invasive procedure. A few mentioned false negative test results and, although only the view of one person, this was raised as a particular issue if the test led to changes to the follow-up pathway and some patients 'slip through the net'.

A few interviewees said it should be the patient's decision to have the test should it be implemented (Memory, attention and decision processes). When explored with other patients although they understood this point of view most thought it impractical if the test were part of routine care and difficult for clinicians not to act particularly if a patient were found to be high risk. Some talked of the shock of receiving the melanoma diagnosis and that further prognostic information at that time may be too much to take in, whereas others thought in practice they both would be reported together.

**What does the test mean for patient care?.** There were two main perceived consequences of the test on patients and their care. These related to the personal impact of more certain risk information and the actions taken based on the test result (Beliefs about consequences).

The consequence of greater certainty about future risk of melanoma recurrence or spread was primarily reassurance and 'peace of mind' both for themselves and their families. This may reflect their experience of risk communication when they were first diagnosed. When this was explored in the interviews most were vague about the details although the majority said they had been informed of the risks. A few recalled the risks were couched in terms of their melanoma grading, that a grade 1 was the 'best situation to be in' (Patient 02). Some said the clinician had explained that they did not know the true risk. In the face of this uncertainty one interviewee said, 'I think then your mind will start wandering and you'll start imagining the worst because that's what we do.' (Patient 08). Therefore, the opportunity to obtain more concrete risk information was generally welcomed.

That the test result would give them some reassurance was based on a low-risk scenario. Conversely, as might be expected, some expressed the view that knowing they were high-risk would be a worry. However, one person said they would prefer to know and then they can 'deal with it' (Patient 16). Awareness that the risk of recurrence or spread was high was said to enable patients to prepare 'You know what you're dealing with' (Patient 01). There was the tendency for a small number of interviewees to interpret a high-risk result as having a bleak outcome, indicating the need to 'put your house in order' and meaning they were 'going to die'.

This second consequence was the subsequent clinical decisions and action on the part of the patient. Based on high-risk test result a small number said one consequence of the test would be for the clinical team to undertake more intense surveillance and recommend treatment. Regarding their own actions in response to a high-risk test result, they stated they would be more vigilant in their self-surveillance and protective behaviours. For patients deemed low risk some thought it could free up clinician time if this group attended less frequently. When frequency of follow-up was explored with other interviewees some said the current level was 'comforting' and others that they would be 'over the moon' if they were told they did not have to attend every three months (Patient 08). Frequent follow-ups did perpetuate concern in certain interviewees and one person also pointed out the impact on their own time, of being seen every three months over the previous three years and having to take time off work to attend appointments. There was awareness of the burden on the health services and even some interviewees who found follow-up reassuring did suggest they could be reduced if the duration of follow up remained as it was currently.

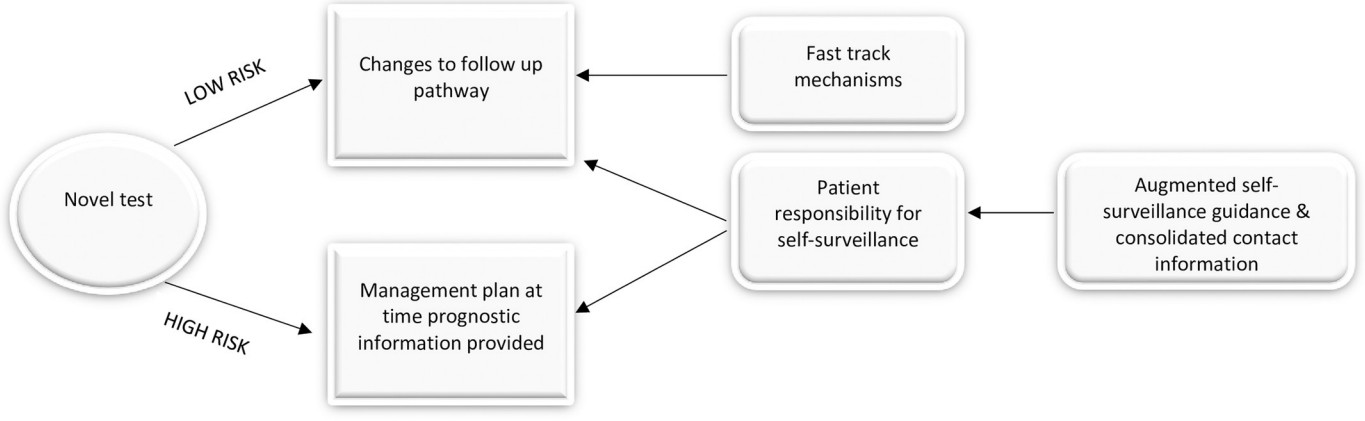

**Fig 3. Mechanisms to support test-based management decisions.**

**Mechanisms and behaviours to support test-based management decisions.** As stated earlier, one consequence of the implementation of the test could be changes to follow-up pathway. However, for these changes to be acceptable interviewees thought the implementation of the novel test should be accompanied by support mechanisms and extra information (Fig 3).

Most thought a change to the follow-up care pathway—either a reduction in the frequency or duration—based on a low risk test result, appropriate if a process was in place to enable quick access to the specialist team should they have any concerns (Behavioural regulation). A fast-track system bypassing the general practitioner, with a phone-call to a specific contact person, was commonly suggested.

Several interviewees talked about their own role in self-surveillance for recurrent melanoma (Social role and identity). Should the test become part of the care-pathway they acknowledged they had a role to play and a responsibility to continue to follow protective measures and regularly check their skin, particularly as one person said, 'Low risk doesn't mean no risk' (Patient15). Some commented on the difficulties in making self-surveillance habitual but thought that particularly if the test classified them as high risk they may be 'a bit more rigorous about doing the self-checking.' (Patient 19).

Although most were confident in checking their skin one person also thought further guidance should be given to patients on what to look out for and training on how to check their lymph nodes (Skills). There were a few comments about providing patients with information packs that contained not only all the relevant contact names and numbers but also guidance on self-surveillance so they could have all the information in one place.

Finally, interviewees thought it crucial that clinicians support the information with immediate and clear management plans. There were comments that patients being told they were high risk without a plan would generate anxiety in patients, even in low grade melanoma. One interviewee said, 'If you're just going to be told you're higher risk but nothing is done about it then it's probably better not to know.' (Patient 10). This relates to the earlier point that some interviewees associated a high-risk test result with a very poor prognosis and reinforces the need for careful communication.

## Discussion

This study identified factors key to the implementation of a novel test from the perspective of health professionals whose potential engagement with the test differed, for example running

the test, conveying the results or using it to plan future care. The test was welcomed by all participants and motivation to adopt such a test was high, particularly in clinician interviewees. Some key facilitators to its implementation were: the clear necessity for a test in the absence of robust prognostic information; the perceived outcomes of tailoring care to patients' needs and the potential to save time and costs for NHS Trusts; and, based on the information provided about the test, histopathologists were confident they had the skills to perform it. Not everyone was comfortable with a change to follow-up based on the test result, and some said this should be the patient's choice. Interviewees disagreed about the potential for changes in follow-up to cause patient anxiety; some said reduced surveillance would support the test information that patients were low risk and decrease anxiety. One recommendation could be to reduce follow-up for those identified as genuinely low risk or follow-up patients up in a primary (GP) rather than secondary care setting.

The main barrier was the resource impact of an additional test on an already overstretched pathology service. However, the histopathologists did not mention this as a barrier. Additionally, interviewees were specific about the requirement for robust data regarding the efficacy of the test. Views were divided as to whether the current validation would be sufficient to convince health practitioners to change practice or if further prospective research was required. NICE approval was mentioned by a few interviewees but the absence of this did not appear to be a barrier to implementation at an individual level.

The need for robust clinical validation of the novel test highlighted by health professional interviewees has led to the expansion of the current validation study to address this concern, increasing both the number and geographic location of the clinical samples. Whilst NICE approval of a medical technology does not mandate use of a particular product over another, it does confirm that the level of clinical and health economic evidence supports adoption by the NHS [15]. Implementation of the novel test will therefore be significantly improved by a positive recommendation from NICE Medical Technology Guidance through the Diagnostic Assessment Programme. Further research is needed regarding the resource implications of an additional test in the pathology workflow and whether provision of additional training would minimise the additional resources required.

Translating research into practice can take 17 years [16]. In this study, exploration of implementation was conducted in parallel with the validation of the novel test rather than sequentially, which reduced the duration of the study. It was a challenge to provide the specific information some health professional interviewees required due to the timing of the qualitative study and the need to protect intellectual property.

From the patient interviewee perspective, the main perceived benefit of the test was to obtain reassurance, but for most this was based on a low-risk test result. Others have reported fear of recurrence to cause the highest anxiety immediately after diagnosis [17] which suggests those with a high-risk test result may need additional psychosocial support as part of their ongoing surveillance. Only one patient interviewee thought the test should be optional and was unsure whether they personally would want to be informed if they were high risk. Research has shown that the majority of melanoma patients wish to have full information about their diagnosis and prognosis [18] but there is very little exploring the psychological impact of prognosis on patients with Stage I/II cutaneous melanoma. There are studies with patients with uveal melanoma and the most recent, a five-year study of 708 survivors with a poor or good prognosis and those who had declined testing, reports 'that harm accruing from a poor prognosis was statistically significant over 5 years but did not exceed general non-cancer population norms.' [19]. The authors conclude that clinicians should inform patients that there may be adverse psychological consequences in the event of a poor prognosis.

There were mixed views as to whether the current frequency of follow-up causes anxiety which is in line with the findings from a systematic review [20]. None of the patient interviewees identified any specific barriers to the implementation of the test but suggested processes and mechanisms that should be in place to support management decisions based on a low- or high-risk result. The mechanism most mentioned was a fast-track process if patients attended less frequently or if the duration of follow-up was reduced. Others have reported that fewer follow-ups were acceptable to some patients if in conjunction with support for skin self-surveillance and rapid clinical review when necessary [21]. Although the potential to be followed up in general practice was not explored in the interviews there were positive and negative accounts of their experiences regarding their melanoma diagnosis. There appeared to be an expectation that the responsibility for follow-up would lie with secondary care. Patients also suggested consolidated contact information and augmented self-surveillance guidance. Some admitted they had not been as vigilant in their self-surveillance as they should, and this is a factor to consider if there are changes to the follow-up pathway.

Understandably some patients could not conceive what a high-risk result meant in terms of management and treatment. Although greater certainty was welcomed, moving from the prognosis currently conveyed of their melanoma being low risk to potentially one of high-risk if the test were implemented created some worry. Both points suggest clinicians will have to consider how they communicate the test result, particularly for those patients whose risk potentially is altered from a low-risk AJCC stage to high-risk with the test.

## Limitations

Histopathologists in particular wanted more details about the test—specifically procedural information such as how long the test would take, which proteins were involved and who would analyse and stain the slides—and some struggled to fully comment on its implementation with the information provided. During recruitment there were histopathologists who declined to participate as they felt there was too little information about the test for them to give an opinion. This led to the inclusion of a lower number of histopathologists than planned and may have resulted in a biased sample in this particular health professional group. Another issue was not providing information beforehand about the validation underway. In hindsight, providing this information earlier would have pre-empted a number of questions about the validation and made the time available for the interview more productive, and may have helped interviewees to better form an opinion about level of evidence.

## Conclusions

In the absence of a robust prognostic test for early-stage melanoma it was unsurprising that health professionals, particularly those who convey risk to patients and determine further management, were positive about the potential consequences of its use. Nevertheless, although widely accepted as a positive contribution to the melanoma prognostic pathway and risk stratification by health professionals, implementation of this novel test would need to consider potential resource implications on pathology services. Clinicians should consider the psychological support that may be required with a high-risk test result. Similarly, patients were supportive of the test and suggested ways in which more certain information regarding melanoma prognosis can be incorporated into the melanoma care pathway.

Exploring implementation of such a novel test at an early stage provided valuable data but there were challenges with health professionals related to the provision of specific details of the test and its validation.

## Supporting information

**S1 File.**

(DOCX)

**S2 File.**

(DOCX)

**S1 Checklist.**

(PDF)

## Acknowledgments

We wish to thank all of the health professionals and patients/carer who took the time to participate in an interview as part of this study and provided valuable insight about the implementation of the novel test.

## Author Contributions

**Conceptualization:** Jan Lecouturier, Marie Labus, Rob A. Ellis, Penny E. Lovat.

**Formal analysis:** Jan Lecouturier, Helen Bosomworth.

**Funding acquisition:** Jan Lecouturier, Marie Labus, Rob A. Ellis, Penny E. Lovat.

**Investigation:** Jan Lecouturier.

**Methodology:** Jan Lecouturier.

**Project administration:** Helen Bosomworth.

**Writing – original draft:** Jan Lecouturier, Helen Bosomworth.

**Writing – review & editing:** Jan Lecouturier, Marie Labus, Rob A. Ellis, Penny E. Lovat.

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
