## [Decision Letter · Decision Letter 0]

12 Oct 2021

PONE-D-21-25124Health professional and patient views of a novel prognostic test for melanoma: a theoretically informed qualitative studyPLOS ONE

Dear Dr. Lecouturier,

Thank you for submitting your manuscript to PLOS ONE. After careful consideration, we feel that it has merit but does not fully meet PLOS ONE’s publication criteria as it currently stands. Therefore, we invite you to submit a revised version of the manuscript that addresses the points raised during the review process.

In accordance with the expert reviewers, I have some minor concerns. Rather than repeat those points here, I refer you to the specific remarks (below) for details.

We look forward to receiving your revised manuscript.

Kind regards,

Nikolas K. Haass, MD/PhD

Academic Editor

PLOS ONE

Journal Requirements:

a) Did participants provide their written or verbal informed consent to participate in this study?

b) If consent was verbal, please explain i) why written consent was not obtained, ii) how you documented participant consent, and iii) whether the ethics committees/IRB approved this consent procedure

3. When reporting the results of qualitative research, we suggest consulting the COREQ guidelines or other relevant checklists listed by the Equator Network, such as the SRQR, to ensure complete reporting (http://journals.plos.org/plosone/s/submission-guidelines#loc-qualitative-research). Moreover, please provide the interview guide used as a Supplementary File

4. Please include your full ethics statement in the ‘Methods’ section of your manuscript file. In your statement, please include the full name of the IRB or ethics committee who approved or waived your study, as well as whether or not you obtained informed written or verbal consent. If consent was waived for your study, please include this information in your statement as well

5. Please include a copy of Table 5 which you refer to in your text on page 8.

6. We note you have included a table to which you do not refer in the text of your manuscript. Please ensure that you refer to Table 3 in your text; if accepted, production will need this reference to link the reader to the Table.

Reviewers' comments:

Reviewer's Responses to Questions

**Comments to the Author**

1. Is the manuscript technically sound, and do the data support the conclusions?

Reviewer #1: Yes

Reviewer #2: Yes

Reviewer #3: Partly

2. Has the statistical analysis been performed appropriately and rigorously? 

Reviewer #1: N/A

Reviewer #2: Yes

Reviewer #3: Yes

3. Have the authors made all data underlying the findings in their manuscript fully available?

Reviewer #1: No

Reviewer #2: Yes

Reviewer #3: No

4. Is the manuscript presented in an intelligible fashion and written in standard English?

Reviewer #1: Yes

Reviewer #2: Yes

Reviewer #3: Yes

5. Review Comments to the Author

Reviewer #1: Thank you for submitting this important exploration of clinician and patient perspectives on new prognostic technology for melanoma. Please see the suggestions below, organised by section:

- ABSTRACT

- The "test" needs to be described, at least briefly. At the moment, all the reader gleans is that it is a "prognostic test"

- INTRODUCTION

- "The AJCC staging criteria are unable to distinguish between truly high and low risk Stage I/II melanomas and currently there are no other consistent ways to accurately determine an individual’s risk of disease progression."

- Current prognostic practices ought to be acknowledged and the marginal, rather than absolute, benefit of this test needs to be measure of interest.

- FINDINGS

- "Very few interviewees had prior knowledge of the novel test other than what was shared as part of the interview."

- Wouldn't it have been better to interview the doctors who had actually been part of the study that used the tests?

- "In contrast, one interviewee did not consider a change to the follow-up pathway a positive outcome of the test. The one-year period of follow-up provided reassurance to the health professional that patients could carry out self-surveillance competently."

- Difficult to understand this sentence

- "One other suggestion was to use the test only where malignancy is detected rather than all suspected melanoma samples"

- Again, even a brief description of the test in this article would be helpful. I assumed that only testing diagnosed melanomas would definitely be the case.

- "Histopathologists in particular wanted more details about the test and some struggled to fully comment on its implementation with the information provided ... During recruitment there were histopathologists who declined to participate as they felt there was too little information about the 25 test for them to give an opinion"

- Related to the point above

- The potential for selection bias here should be acknowledged - it would be reasonable to expect that those who felt they had insufficient information about the test to participate may have given comparatively less enthusiastic answers were they included

- MISCELLANEOUS

- It would be good to know some of the test parameters - the predictive values, particularly

- The hierarchy of headings in the article isn't quite clear - to my understanding, "Implementation" belongs under "Health professional views of the novel prognostic test", however, according to the text styling, appears to be at the same level of the hierarchy. Further, this seems to be at the same level as Findings, which they would belong under. Also note the discrepancy in phrasing of "Health professional views of the novel prognostic test" and "Patients views of the novel test"

- REFERENCES

- Needs formatting

- FIGURES AND TABLES

- Table 1

- Is reference 13 the citation for this? Perhaps that should be acknowledged in the table caption (as well as the main text)

- Figure 3

- "Management plan at time prognostic information provided"

- Unclear meaning

- Suggest it be redesigned somewhat. Perhaps all arrows point to the right, and consider removing "Patient responsibility for self-surveillance"

- As, for example, the "Fast track mechanisms" are not explained in the figure (and it might be assumed that it would be more appropriate for the high risk, rather than low risk patients), reconsider whether this figure is important to include. It portrays changes to management that might result from the implementation of the test, but is not really portraying a clinical pathway, and therefore the flowchart format isn't especially illuminating.

Reviewer #2: I congratulate the authors on their work. The article reports a very interesting study that addresses a delicate and extremely important issue in the current context. The advent of predictive prognostic algorithms is also affecting other tumor sites or other pathologies, such as amyotrophic lateral scllorosis and knowing the opinion of clinicians and patients is essential before proceeding with the adoption of the tool within the care pathways.

In my opinion, the article is well written and structured, the methodology seems adequate and well described (I admit that I do not have a thorough knowledge of qualitative techniques and do not use them habitually), the results are clearly presented and also the discussion of it seems well done. I have no substantial criticism to make, if not a few comments that do not affect my positive opinion on the article.

The health professionals were contacted by email; it is possible that many have refused. The authors do not provide any data in this regard. It would be interesting to report the numbers and reasons for the refusal. It may be possible that those who have accepted, even if they do not know the test previously, are still professionals generally in favor of the new, with a flexible and accommodating attitude towards changes and well disposed towards confronting patients and communicating with them. In that sense, we would have a sampling bias. Some clinicians may be frankly opposed, others in favor of the test itself, but they may fear that they do not have personal or team resources to manage any negative psychological reactions of patients. Authors should at least argue more about this aspect for discussion and at best provide more data on rejecting subjects.

In addition to this I point out that tables are numbered incorrectly.

Reviewer #3: This paper provides a qualitative analysis of likely implementation issues regarding a prognostic test for metastatic spread of cutaneous melanoma. The analysis is informed by the Theoretical Domains Framework. Findings suggest that 20 practitioners, 19 patients and a carer were broadly supportive (sometimes enthusiastic) of the test; pending demonstration of specificity and sensitivity and the influence of the test on care pathways for practitioners, and concerns about the influence of the test on care pathways and self-surveillance.

My main comment about the work pertains to how it fits into the broader social science literature surrounding prognostication. Indeed, the authors do not seem to mention this at all. First, most work on prognostication is performed against the background of a test of known sensitivity and specificity. Here, these features are unknown, and thus the comments are based on a hypothetical rather than real scenario. This carries two problems; that participants may feel differently when they face implementation issues for a real test, and that responses may well differ according to how accurate the test is (this was prominently raised by practitioners in this study, and indeed some histopathologists declined to participate because they lacked information about the test).

Second, the authors make an explicit choice to focus on implementation issues. However, most of the prognostication literature raises ethical issues around whether the test should be used at all and safeguards against potential harm if it is used. Highly accurate prognostic testing in Huntington’s disease and uveal melanoma has shown that a prognosis of reduced life expectancy is associated with potential psychological harm for patients or at least regret that they chose prognostic testing. The implications for this current test are unclear because its accuracy is not yet known, but despite ethical issues being prominent in the literature they are not acknowledged in this paper. I think that, before implementation issues are considered, there is a need to acknowledge ethical issues associated with prognostication, to argue that the test is ethically justifiable and to discuss safeguards in both making the offer of the test and helping patients to deal with the consequences.

In sum, I find the paper to be lacking in important areas and cannot yet see its place in the prognostication literature. I imagine that these problems can be remediated with the data available to the researchers (although the analysis may need to move into areas not covered by the TDF) and by linking the paper more explicitly to the prognostication literature.

6. PLOS authors have the option to publish the peer review history of their article (what does this mean?). If published, this will include your full peer review and any attached files.

Reviewer #1: No

Reviewer #2: **Yes: **Dr. Marco Miniotti, PhD

Reviewer #3: No

---

## [Author Response · Author response to Decision Letter 0]

16 Nov 2021

1. We have checked to ensure our manuscript meets PLOS ONE's style requirements. 

2. We have amended our ethics statement to address the provision of verbal consent, why written consent was not obtained, how consent was documented, and that this was approved by the ethics committee.

3. We have completed and uploaded the COREQ checklists and the two topic guides as supplementary 

4. We have provided a full ethics statement in the ‘Methods’ section including the full name of the ethics committee who approved the study, including details of verbal consent. 

5. The reference to Table 5 was an error. We have amended the text regarding Table 3. 

7. We have reviewed our reference list and are unaware that any included papers that have been retracted. 

Response to reviewers

Reviewer #1: Thank you for submitting this important exploration of clinician and patient perspectives on new prognostic technology for melanoma. Please see the suggestions below, organised by section:

- ABSTRACT

- The "test" needs to be described, at least briefly. At the moment, all the reader gleans is that it is a "prognostic test"

We have added a description of the test in the Abstract (LINES 44-7).

- INTRODUCTION

- "The AJCC staging criteria are unable to distinguish between truly high and low risk Stage I/II melanomas and currently there are no other consistent ways to accurately determine an individual’s risk of disease progression."

- Current prognostic practices ought to be acknowledged and the marginal, rather than absolute, benefit of this test needs to be measure of interest.

We are a little unsure what the reviewer is seeking here but have reiterated the ability of the current version of the AJCC staging criteria to categorise tumours as stage I or II while highlighting the inability of these criteria to distinguish between at risk and low risk tumour subsets (LINES 111-12). 

- FINDINGS

- "Very few interviewees had prior knowledge of the novel test other than what was shared as part of the interview."

- Wouldn't it have been better to interview the doctors who had actually been part of the study that used the tests?

At the time of the present study the novel test described within our manuscript was undergoing further clinical validation. The test was not being used in any of the hospital trusts where the interviewees were based. If interviewees had prior knowledge of the novel test this was likely to be from a presentation given by one of the co-authors, about the development of the test at national/international meetings (including the British Society for Investigative Dermatology and Melanoma Focus) or from the paper published paper detailing results of the test in discovery cohorts of AJCC stage I melanoma (Ellis et al 2020) a reference included in the present manuscript. We have reinforced this within the manuscript (LINES 261-63). 

- "In contrast, one interviewee did not consider a change to the follow-up pathway a positive outcome of the test. The one-year period of follow-up provided reassurance to the health professional that patients could carry out self-surveillance competently."

- Difficult to understand this sentence

We have amended the sentence to make this point clearer (LINES 303-05).

- "One other suggestion was to use the test only where malignancy is detected rather than all suspected melanoma samples"

- Again, even a brief description of the test in this article would be helpful. I assumed that only testing diagnosed melanomas would definitely be the case.

"prognostic test"

We have added a description of the test in the main body of the manuscript (LINES 130-40).

- "Histopathologists in particular wanted more details about the test and some struggled to fully comment on its implementation with the information provided ... During recruitment there were histopathologists who declined to participate as they felt there was too little information about the 25 test for them to give an opinion"

- Related to the point above

- The potential for selection bias here should be acknowledged - it would be reasonable to expect that those who felt they had insufficient information about the test to participate may have given comparatively less enthusiastic answers were they included

We have expanded upon this in the Limitations section (LINES 602-8). 

- MISCELLANEOUS

- It would be good to know some of the test parameters - the predictive values, particularly

We have added a sentence describing some of the test parameters in a discovery cohort (LINES 134-6)

- The hierarchy of headings in the article isn't quite clear - to my understanding, "Implementation" belongs under "Health professional views of the novel prognostic test", however, according to the text styling, appears to be at the same level of the hierarchy. Further, this seems to be at the same level as Findings, which they would belong under. Also note the discrepancy in phrasing of "Health professional views of the novel prognostic test" and "Patients views of the novel test"

Thank you for pointing out the error in the text styling and discrepancy in phrasing which we have now amended.

- REFERENCES

- Needs formatting

The references have been formatted.

- FIGURES AND TABLES

- Table 1

- Is reference 13 the citation for this? Perhaps that should be acknowledged in the table caption (as well as the main text)

The reference was cited in the main text ‘A thematic [13] analysis was conducted which involved a process of familiarisation with the data to develop health professional and patient coding frameworks (Table 1)’ but we have now moved the reference number to the end of this sentence to link the table to the reference more clearly (LINE 222). We have now acknowledged reference 13 in the table caption (LINE 229)

- Figure 3

- "Management plan at time prognostic information provided"

- Unclear meaning

- Suggest it be redesigned somewhat. Perhaps all arrows point to the right, and consider removing "Patient responsibility for self-surveillance"

- As, for example, the "Fast track mechanisms" are not explained in the figure (and it might be assumed that it would be more appropriate for the high risk, rather than low risk patients), reconsider whether this figure is important to include. It portrays changes to management that might result from the implementation of the test, but is not really portraying a clinical pathway, and therefore the flowchart format isn't especially illuminating.

We would like to thank Reviewer 2 for the detailed feedback on Figure 3. We have amended the text to ensure it is clear to the reader without referring to the text. Patient interviewees suggested the fast-track mechanisms for those with a low-risk test result should their follow-up pathway be less frequent or shortened. It was not mentioned as an option for high-risk patients, as the assumption was they would be followed up regularly and have increased clinical input. We hope the revised figure is clearer for the reader

Reviewer #2: I congratulate the authors on their work. The article reports a very interesting study that addresses a delicate and extremely important issue in the current context. The advent of predictive prognostic algorithms is also affecting other tumor sites or other pathologies, such as amyotrophic lateral scllorosis and knowing the opinion of clinicians and patients is essential before proceeding with the adoption of the tool within the care pathways.

In my opinion, the article is well written and structured, the methodology seems adequate and well described (I admit that I do not have a thorough knowledge of qualitative techniques and do not use them habitually), the results are clearly presented and also the discussion of it seems well done. I have no substantial criticism to make, if not a few comments that do not affect my positive opinion on the article.

We thank reviewer 2 for the very positive comments about this article.

1. The health professionals were contacted by email; it is possible that many have refused. The authors do not provide any data in this regard. It would be interesting to report the numbers and reasons for the refusal. It may be possible that those who have accepted, even if they do not know the test previously, are still professionals generally in favor of the new, with a flexible and accommodating attitude towards changes and well disposed towards confronting patients and communicating with them. In that sense, we would have a sampling bias. Some clinicians may be frankly opposed, others in favor of the test itself, but they may fear that they do not have personal or team resources to manage any negative psychological reactions of patients. Authors should at least argue more about this aspect for discussion and at best provide more data on rejecting subjects.

In response to the comment about the numbers approached and refusals it is probably helpful to explain in detail the recruitment pathway. Information about the study calling for health professionals to participate was circulated through two professional organisations/groups. Names were passed to the researcher or those who were interested could contact the researcher directly to arrange an interview. Anyone connected to the study or the development of the novel test was excluded.

The first organisation was Melanoma Focus https://melanomafocus.org/members/apply-or-pay-for-membership/ It is not possible to state how many Melanoma Focus professional members would have received study information and therefore it is difficult to determine a response rate or reasons for not responding. 

The second organisation was the Northern Cancer Alliance Skin Cancer Speciality Group https://northerncanceralliance.nhs.uk/advisory_group/skin-expert-advisory-group/ (the lead of which is a co-applicant on the study). Members were asked to cascade the information to relevant others outside of the northern region. Again, it is difficult to determine the total number who received study information and overall response rate.

We have provided more detail about the recruitment process for health professionals (LINES 176-85) and hope this will address these important points.

2. In addition to this I point out that tables are numbered incorrectly.

We have amended the table numbers.

Reviewer #3: This paper provides a qualitative analysis of likely implementation issues regarding a prognostic test for metastatic spread of cutaneous melanoma. The analysis is informed by the Theoretical Domains Framework. Findings suggest that 20 practitioners, 19 patients and a carer were broadly supportive (sometimes enthusiastic) of the test; pending demonstration of specificity and sensitivity and the influence of the test on care pathways for practitioners, and concerns about the influence of the test on care pathways and self-surveillance.

1. My main comment about the work pertains to how it fits into the broader social science literature surrounding prognostication. Indeed, the authors do not seem to mention this at all. First, most work on prognostication is performed against the background of a test of known sensitivity and specificity. Here, these features are unknown, and thus the comments are based on a hypothetical rather than real scenario. This carries two problems; that participants may feel differently when they face implementation issues for a real test, and that responses may well differ according to how accurate the test is (this was prominently raised by practitioners in this study, and indeed some histopathologists declined to participate because they lacked information about the test).

We believed that health professionals had the requisite experience and understanding of the care pathway to be able to provide valid comments on barriers/enablers of implementing this test despite the fact it was not currently used in clinical practice. We expected test accuracy and the need for evidence to be cited as an important factor. However, we were interested in other aspects that would impact on its adoption, such as what level of evidence they required and factors related to the melanoma care pathway, for example burden on pathology services. We have added a sentence to the Methods section (LINES 202-4).

Also, just to add, the main information needs of the histopathologists who declined were procedural issues - how long the test would take, which proteins were involved, who would analyse and stain the slides - rather than about the accuracy of the test. A sentence has been added to the Discussion to reflect this (LINES 602-4)

2. Second, the authors make an explicit choice to focus on implementation issues. However, most of the prognostication literature raises ethical issues around whether the test should be used at all and safeguards against potential harm if it is used. Highly accurate prognostic testing in Huntington’s disease and uveal melanoma has shown that a prognosis of reduced life expectancy is associated with potential psychological harm for patients or at least regret that they chose prognostic testing. The implications for this current test are unclear because its accuracy is not yet known, but despite ethical issues being prominent in the literature they are not acknowledged in this paper. I think that, before implementation issues are considered, there is a need to acknowledge ethical issues associated with prognostication, to argue that the test is ethically justifiable and to discuss safeguards in both making the offer of the test and helping patients to deal with the consequences.

We thank Reviewer 3 for raising this point but in defence of our paper we believe it sits within the implementation rather than prognostication literature as the key objective was to explore the barriers and enablers to the implementation of the novel test into the melanoma care pathway. This is described in the Introduction (LINES 146-9). 

We agree that there can be ethical issues around the use of prognostic tests. However, in this specific patient group the current prognostic method is sub-optimal as it does not determine those truly high or low risk which is also problematic. As demonstrated in the paper this uncertainty was an issue raised by both health professional and patient interviewees. 

Regarding the test, one patient interviewee questioned whether they themselves would want to know if they were high risk and in subsequent interviews others were asked whether it should be optional. The majority view was that more certain information about prognosis would simply become part of the usual discussion about future risk that occurs in the clinician-patient consultation. In light of this finding and the reviewer comments we have referenced the most up to date literature about the psychological impact of prognostic information on patients with uveal melanoma but were unable to find research conducted with a cutaneous melanoma population (LINES 569-78)

3. In sum, I find the paper to be lacking in important areas and cannot yet see its place in the prognostication literature. I imagine that these problems can be remediated with the data available to the researchers (although the analysis may need to move into areas not covered by the TDF) and by linking the paper more explicitly to the prognostication literature.

We hope that the responses and amendments to the manuscript as described above will address the points made by Reviewer 3.

---

## [Decision Letter · Decision Letter 1]

23 Feb 2022

Health professional and patient views of a novel prognostic test for melanoma: a theoretically informed qualitative study

PONE-D-21-25124R1

Dear Dr. Lecouturier,

We’re pleased to inform you that your manuscript has been judged scientifically suitable for publication and will be formally accepted for publication once it meets all outstanding technical requirements.

Within one week, you’ll receive an e-mail detailing the required amendments. When these have been addressed, you’ll receive a formal acceptance letter and your manuscript will be scheduled for publication. Please also discuss the first reviewer's suggestions with your team and edit the manuscript according to the outcome of your discussion.

Kind regards,

Nikolas K. Haass, MD/PhD

Academic Editor

PLOS ONE

Additional Editor Comments (optional):

Please also discuss the first reviewer's suggestions with your team and edit the manuscript according to the outcome of your discussion.

Reviewers' comments:

Reviewer's Responses to Questions

**Comments to the Author**

1. If the authors have adequately addressed your comments raised in a previous round of review and you feel that this manuscript is now acceptable for publication, you may indicate that here to bypass the “Comments to the Author” section, enter your conflict of interest statement in the “Confidential to Editor” section, and submit your "Accept" recommendation.

Reviewer #1: (No Response)

Reviewer #2: All comments have been addressed

2. Is the manuscript technically sound, and do the data support the conclusions?

Reviewer #1: Yes

Reviewer #2: Yes

3. Has the statistical analysis been performed appropriately and rigorously? 

Reviewer #1: N/A

Reviewer #2: Yes

4. Have the authors made all data underlying the findings in their manuscript fully available?

Reviewer #1: No

Reviewer #2: Yes

5. Is the manuscript presented in an intelligible fashion and written in standard English?

Reviewer #1: Yes

Reviewer #2: Yes

6. Review Comments to the Author

Reviewer #1: Thank you for the amendments. I have some comments:

Suggest "prediction of prognosis" be replaced by either "prognosis" or "prognostication". Probably simply "prognosis".

Line 112: "current" should be "currently" (or remove the following "the")

I think the references to line numbers has gotten off kilter.

"At the time of the present study the novel test described within our manuscript was undergoing further clinical validation. The test was not being used in any of the hospital trusts where the interviewees were based. If interviewees had prior knowledge of the novel test this was likely to be from a presentation given by one of the co-authors, about the development of the test at national/international meetings (including the British Society for Investigative Dermatology and Melanoma Focus) or from the paper published paper detailing results of the test in discovery cohorts of AJCC stage I melanoma (Ellis et al 2020) a reference included in the present manuscript. We have reinforced this within the manuscript (LINES 261-63)."

- I recommend that it be stated in the text that "At the time of the present study the novel test described within our manuscript was undergoing further clinical validation"

"underpinning the lack for any additional clinical procedures required for the patient." - Suggest "highlighting" or "emphasising" rather than "underpinning" here.

"We have added a description of the test in the main body of the manuscript (LINES 130-40)."

- I think perhaps this could be rephrased to avoid the implication that it was ever considered that non-melanomas. Perhaps "In accordance with the intended use, one other suggestion was to..."

"We have expanded upon this [sampling bias] in the Limitations section (LINES 602-8).

- I suggest it be specified that this phenomenon would likely have caused a bias of positive sentiment.

"The ethics committee agreed this procedure." Missing a "to"

Figure 3 still seems not totally clear to me. I believe a strong case could be made that it not be included, as the description in the text is adequate.

Nevertheless, it would seem to me to be more intuitive that, as portrayed, the novel test and then its outcomes sit at the beginning of the flowchart, and then the implications that follow on from the results are represented downstream of those results in a stepwise, one-way manner.

As I understand it, the "Changes to follow up pathway" are in fact the "Fast track mechanisms", and hence these should be represented within the same cell, such as "Changes to follow up pathway (fast track mechanisms")

Similarly, "Patient responsibility for self-surveillance" and "Augmented self-surveillance guidance & consolidated contact information" refer to things that occur at the same step in the pathway and should be represented in the same cell. Perhaps "Augmented self-surveillance guidance and patient responsibility for self-surveillance, with consolidated contact information". It would also seem more intuitive to me that the arrows point towards the right.

Reviewer #2: The authors have adequately addressed the comments I've raised up. I have not further comments or suggestions to provide them.

7. PLOS authors have the option to publish the peer review history of their article (what does this mean?). If published, this will include your full peer review and any attached files.

Reviewer #1: No

Reviewer #2: **Yes: **Dr. Marco Miniotti

---

## [Editor Report · Acceptance letter]

25 Mar 2022

PONE-D-21-25124R1 

Health professional and patient views of a novel prognostic test for melanoma: a theoretically informed qualitative study 

Dear Dr. Lecouturier:

I'm pleased to inform you that your manuscript has been deemed suitable for publication in PLOS ONE. Congratulations! Your manuscript is now with our production department. 

Kind regards, 

on behalf of

Prof Nikolas K. Haass 

Academic Editor

PLOS ONE